# Sign Cauchy Projections and Chi-Square Kernel

**Ping Li**
Dept of Statistics & Biostat.
Dept of Computer Science
Rutgers University
pingli@stat.rutgers.edu

**Gennady Samorodnitsky**
ORIE and Dept of Stat. Science
Cornell University
Ithaca, NY 14853
gs18@cornell.edu

**John Hopcroft**
Dept of Computer Science
Cornell University
Ithaca, NY 14853
jeh@cs.cornell.edu

## Abstract

The method of *stable random projections* is useful for efficiently approximating the $l_\alpha$ distance ($0 < \alpha \le 2$) in high dimension and it is naturally suitable for data streams. In this paper, we propose to use only the signs of the projected data and we analyze the probability of collision (i.e., when the two signs differ). Interestingly, when $\alpha = 1$ (i.e., Cauchy random projections), we show that the probability of collision can be accurately approximated as functions of the chi-square ($\chi^2$) similarity. In text and vision applications, the $\chi^2$ similarity is a popular measure when the features are generated from histograms (which are a typical example of data streams). Experiments confirm that the proposed method is promising for large-scale learning applications. The full paper is available at ***arXiv:1308.1009***.

There are many future research problems. For example, when $\alpha \to 0$, the collision probability is a function of the resemblance (of the binary-quantized data). This provides an effective mechanism for resemblance estimation in data streams.

## 1 Introduction

High-dimensional representations have become very popular in modern applications of machine learning, computer vision, and information retrieval. For example, Winner of 2009 PASCAL image classification challenge used millions of features [29]. [1, 30] described applications with billion or trillion features. The use of high-dimensional data often achieves good accuracies at the cost of a significant increase in computations, storage, and energy consumptions.

Consider two data vectors (e.g., two images) $u, v \in \mathbb{R}^D$. A basic task is to compute their *distance* or *similarity*. For example, the correlation ($\rho_2$) and $l_\alpha$ distance ($d_\alpha$) are commonly used:

$$\rho_2(u,v) = \frac{\sum_{i=1}^{D} u_i v_i}{\sqrt{\sum_{i=1}^{D} u_i^2 \sum_{i=1}^{D} v_i^2}}, \qquad d_\alpha(u,v) = \sum_{i=1}^{D} |u_i - v_i|^\alpha \qquad (1)$$

In this study, we are particularly interested in the $\chi^2$ similarity, denoted by $\rho_{\chi^2}$:

$$\rho_{\chi^2} = \sum_{i=1}^{D} \frac{2u_i v_i}{u_i + v_i}, \quad \text{where} \quad u_i \ge 0, \ v_i \ge 0, \ \sum_{i=1}^{D} u_i = \sum_{i=1}^{D} v_i = 1 \qquad (2)$$

The chi-square similarity is closely related to the chi-square distance $d_{\chi^2}$:

$$d_{\chi^2} = \sum_{i=1}^{D} \frac{(u_i - v_i)^2}{u_i + v_i} = \sum_{i=1}^{D} (u_i + v_i) - \sum_{i=1}^{D} \frac{4u_i v_i}{u_i + v_i} = 2 - 2\rho_{\chi^2} \qquad (3)$$

The chi-square similarity is an instance of the Hilbertian metrics, which are defined over probability space [10] and suitable for data generated from histograms. Histogram-based features (e.g., bag-of-word or bag-of-visual-word models) are extremely popular in computer vision, natural language processing (NLP), and information retrieval. Empirical studies have demonstrated the superiority of the $\chi^2$ distance over $l_2$ or $l_1$ distances for image and text classification tasks [4, 10, 13, 2, 28, 27, 26].

The method of *normal random projections* (i.e., $\alpha$-stable projections with $\alpha = 2$) has become popular in machine learning (e.g., [7]) for reducing the data dimensions and data sizes, to facilitate

efficient computations of the $l_2$ distances and correlations. More generally, the method of *stable random projections* [11, 17] provides an efficient algorithm to compute the $l_\alpha$ distances ($0 < \alpha \leq 2$). In this paper, we propose to use only the signs of the projected data after applying stable projections.

## 1.1 Stable Random Projections and Sign (1-Bit) Stable Random Projections

Consider two high-dimensional data vectors $u, v \in \mathbb{R}^D$. The basic idea of stable random projections is to multiply $u$ and $v$ by a random matrix $\mathbf{R} \in \mathbb{R}^{D \times k}$: $x = u\mathbf{R} \in \mathbb{R}^k$, $y = v\mathbf{R} \in \mathbb{R}^k$, where entries of $\mathbf{R}$ are i.i.d. samples from a symmetric $\alpha$-stable distribution with unit scale. By properties of stable distributions, $x_j - y_j$ follows a symmetric $\alpha$-stable distribution with scale $d_\alpha$. Hence, the task of computing $d_\alpha$ boils down to estimating the scale $d_\alpha$ from $k$ i.i.d. samples. In this paper, we propose to store only the signs of projected data and we study the probability of collision:

$$P_\alpha = \mathbf{Pr}\left(\text{sign}(x_j) \neq \text{sign}(y_j)\right) \tag{4}$$

Using only the signs (i.e., 1 bit) has significant advantages for applications in search and learning. When $\alpha = 2$, this probability can be analytically evaluated [9] (or via a simple geometric argument):

$$P_2 = \mathbf{Pr}\left(\text{sign}(x_j) \neq \text{sign}(y_j)\right) = \frac{1}{\pi} \cos^{-1} \rho_2 \tag{5}$$

which is an important result known as *sim-hash* [5]. For $\alpha < 2$, the collision probability is an open problem. When the data are nonnegative, this paper (Theorem 1) will prove a bound of $P_\alpha$ for general $0 < \alpha \leq 2$. The bound is exact at $\alpha = 2$ and becomes less sharp as $\alpha$ moves away from 2. Furthermore, for $\alpha = 1$ and nonnegative data, we have the interesting observation that the probability $P_1$ can be well approximated as functions of the $\chi^2$ similarity $\rho_{\chi^2}$.

## 1.2 The Advantages of Sign Stable Random Projections

1. There is a significant saving in storage space by using only 1 bit instead of (e.g.,) 64 bits.

2. This scheme leads to an efficient linear algorithm (e.g., linear SVM). For example, a negative sign can be coded as "01" and a positive sign as "10" (i.e., a vector of length 2). With $k$ projections, we concatenate $k$ short vectors to form a vector of length $2k$. This idea is inspired by *b-bit minwise hashing* [20], which was designed for binary sparse data.

3. This scheme also leads to an efficient near neighbor search algorithm [8, 12]. We can code a negative sign by "0" and positive sign by "1" and concatenate $k$ such bits to form a hash table of $2^k$ buckets. In the query phase, one only searches for similar vectors in one bucket.

## 1.3 Data Stream Computations

Stable random projections are naturally suitable for *data streams*. In modern applications, massive datasets are often generated in a streaming fashion, which are difficult to transmit and store [22], as the processing is done on the fly in one-pass of the data. In the standard *turnstile* model [22], a data stream can be viewed as high-dimensional vector with the entry values changing over time.

Here, we denote a stream at time $t$ by $u_i^{(t)}$, $i = 1$ to $D$. At time $t$, a stream element $(i_t, I_t)$ arrives and updates the $i_t$-th coordinate as $u_{i_t}^{(t)} = u_{i_t}^{(t-1)} + I_t$. Clearly, the turnstile data stream model is particularly suitable for describing histograms and it is also a standard model for network traffic summarization and monitoring [31]. Because this stream model is linear, methods based on linear projections (i.e., matrix-vector multiplications) can naturally handle streaming data of this sort. Basically, entries of the projection matrix $\mathbf{R} \in \mathbb{R}^{D \times k}$ are (re)generated as needed using pseudo-random number techniques [23]. As $(i_t, I_t)$ arrives, only the entries in the $i_t$-th row, i.e., $r_{i_t,j}, j = 1$ to $k$, are (re)generated and the projected data are updated as $x_j^{(t)} = x_j^{(t-1)} + I_t \times r_{i_t j}$.

Recall that, in the definition of $\chi^2$ similarity, the data are assumed to be normalized (summing to 1). For nonnegative streams, the sum can be computed error-free by using merely one counter: $\sum_{i=1}^{D} u_i^{(t)} = \sum_{s=1}^{t} I_s$. Thus we can still use, without loss of generality, the sum-to-one assumption, even in the streaming environment. This fact was recently exploited by another data stream algorithm named *Compressed Counting (CC)* [18] for estimating the *Shannon entropy* of streams.

Because the use of the $\chi^2$ similarity is popular in (e.g.,) computer vision, recently there are other proposals for estimating the $\chi^2$ similarity. For example, [15] proposed a nice technique to approximate $\rho_{\chi^2}$ by first expanding the data from $D$ dimensions to (e.g.,) $5 \sim 10 \times D$ dimensions through a *nonlinear* transformation and then applying normal random projections on the expanded data. The nonlinear transformation makes their method not applicable to data streams, unlike our proposal.

For notational simplicity, we will drop the superscript $(t)$ for the rest of the paper.

## 2 An Experimental Study of Chi-Square Kernels

We provide an experimental study to validate the use of $\chi^2$ similarity. Here, the "$\chi^2$-kernel" is defined as $K(u,v) = \rho_{\chi^2}$ and the "acos-$\chi^2$-kernel" as $K(u,v) = 1 - \frac{1}{\pi}\cos^{-1}\rho_{\chi^2}$. With a slight abuse of terminology, we call both "$\chi^2$ kernel" when it is clear in the context.

We use the "precomputed kernel" functionality in LIBSVM on two datasets: (i) *UCI-PEMS*, with 267 training examples and 173 testing examples in 138,672 dimensions; (ii) *MNIST-small*, a subset of the popular *MNIST* dataset, with 10,000 training examples and 10,000 testing examples.

The results are shown in Figure 1. To compare these two types of $\chi^2$ kernels with "linear" kernel, we also test the same data using LIBLINEAR [6] after normalizing the data to have unit Euclidian norm, i.e., we basically use $\rho_2$. For both LIBSVM and LIBLINEAR, we use $l_2$-regularization with a regularization parameter $C$ and we report the test errors for a wide range of $C$ values.

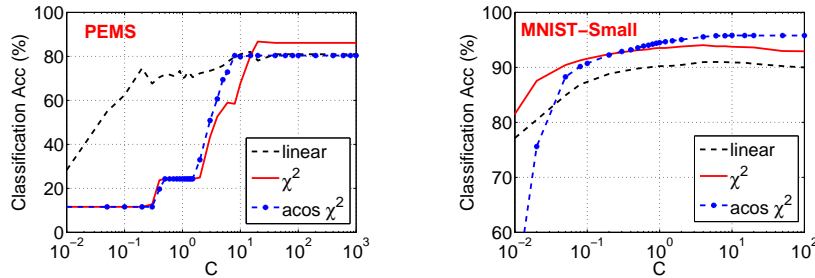

Figure 1: Classification accuracies. $C$ is the $l_2$-regularization parameter. We use LIBLINEAR for "linear" (i.e., $\rho_2$) kernel and LIBSVM "precomputed kernel" for two types of $\chi^2$ kernels ("$\chi^2$-kernel" and "acos-$\chi^2$-kernel"). For *UCI-PEMS*, the $\chi^2$-kernel has better performance than the linear kernel and acos-$\chi^2$-kernel. For *MNIST-Small*, both $\chi^2$ kernels noticeably outperform linear kernel. Note that *MNIST-small* used the original MNIST test set and merely 1/6 of the original training set.

Here, we should state that it is not the intention of this paper to use these two small examples to conclude the advantage of $\chi^2$ kernels over linear kernel. We simply use them to validate our proposed method, which is general-purpose and is not limited to data generated from histograms.

## 3 Sign Stable Random Projections and the Collision Probability Bound

We apply stable random projections on two vectors $u,v \in \mathbb{R}^D$: $x = \sum_{i=1}^{D} u_i r_i$, $y = \sum_{i=1}^{D} v_i r_i$, $r_i \sim S(\alpha, 1)$, i.i.d. Here $Z \sim S(\alpha, \gamma)$ denotes a symmetric $\alpha$-stable distribution with scale $\gamma$, whose characteristic function [24] is $E\left(e^{\sqrt{-1}Zt}\right) = e^{-\gamma|t|^\alpha}$. By properties of stable distributions, we know $x - y \sim S\left(\alpha, \sum_{i=1}^{D}|u_i - v_i|^\alpha\right)$. Applications including linear learning and near neighbor search will benefit from *sign $\alpha$-stable random projections*. When $\alpha = 2$ (i.e. normal), the collision probability $\mathbf{Pr}\left(\text{sign}(x) \neq \text{sign}(y)\right)$ is known [5, 9]. For $\alpha < 2$, it is a difficult probability problem. This section provides a bound of $\mathbf{Pr}\left(\text{sign}(x) \neq \text{sign}(y)\right)$, which is fairly accurate for $\alpha$ close to 2.

### 3.1 Collision Probability Bound

In this paper, we focus on nonnegative data (as common in practice). We present our first theorem.

**Theorem 1** *When the data are nonnegative, i.e., $u_i \geq 0, v_i \geq 0$, we have*

$$\mathbf{Pr}\left(sign(x) \neq sign(y)\right) \leq \frac{1}{\pi}\cos^{-1}\rho_\alpha, \quad where \ \rho_\alpha = \left(\frac{\sum_{i=1}^{D} u_i^{\alpha/2} v_i^{\alpha/2}}{\sqrt{\sum_{i=1}^{D} u_i^\alpha \sum_{i=1}^{D} v_i^\alpha}}\right)^{2/\alpha} \quad \Box \quad (6)$$

For $\alpha = 2$, this bound is exact [5, 9]. In fact the result for $\alpha = 2$ leads to the following Lemma:

**Lemma 1** *The kernel defined as $K(u,v) = 1 - \frac{1}{\pi}\cos^{-1}\rho_2$ is positive definite (PD).*

**Proof:** *The indicator function $1\{sign(x) = sign(y)\}$ can be written as an inner product (hence PD) and $\mathbf{Pr}\left(sign(x) = sign(y)\right) = E\left(1\{sign(x) = sign(y)\}\right) = 1 - \frac{1}{\pi}\cos^{-1}\rho_2$.* $\Box$

## 3.2 A Simulation Study to Verify the Bound of the Collision Probability

We generate the original data $u$ and $v$ by sampling from a bivariate t-distribution, which has two parameters: the correlation and the number of degrees of freedom (which is taken to be 1 in our experiments). We use a full range of the correlation parameter from 0 to 1 (spaced at 0.01). To generate positive data, we simply take the absolute values of the generated data. Then we fix the data as our original data (like $u$ and $v$), apply sign stable random projections, and report the empirical collision probabilities (after $10^5$ repetitions).

Figure 2 presents the simulated collision probability $\mathbf{Pr}\left(\text{sign}(x) \neq \text{sign}(y)\right)$ for $D = 100$ and $\alpha \in \{1.5, 1.2, 1.0, 0.5\}$. In each panel, the dashed curve is the theoretical upper bound $\frac{1}{\pi}\cos^{-1}\rho_\alpha$, and the solid curve is the simulated collision probability. Note that it is expected that the simulated data can not cover the entire range of $\rho_\alpha$ values, especially as $\alpha \to 0$.

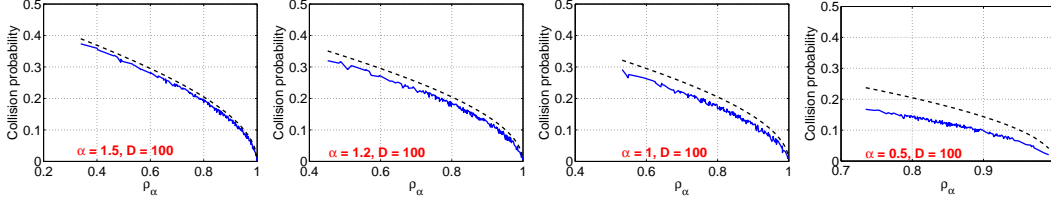

Figure 2: **Dense Data** and $D = 100$. Simulated collision probability $\mathbf{Pr}\left(\text{sign}(x) \neq \text{sign}(y)\right)$ for sign stable random projections. In each panel, the dashed curve is the upper bound $\frac{1}{\pi}\cos^{-1}\rho_\alpha$.

Figure 2 verifies the theoretical upper bound $\frac{1}{\pi}\cos^{-1}\rho_\alpha$. When $\alpha \geq 1.5$, this upper bound is fairly sharp. However, when $\alpha \leq 1$, the bound is not tight, especially for small $\alpha$. Also, the curves of the empirical collision probabilities are not smooth (in terms of $\rho_\alpha$).

Real-world high-dimensional datasets are often **sparse**. To verify the theoretical upper bound of the collision probability on sparse data, we also simulate sparse data by randomly making $50\%$ of the generated data as used in Figure 2 be zero. With sparse data, it is even more obvious that the theoretical upper bound $\frac{1}{\pi}\cos^{-1}\rho_\alpha$ is not sharp when $\alpha \leq 1$, as shown in Figure 3.

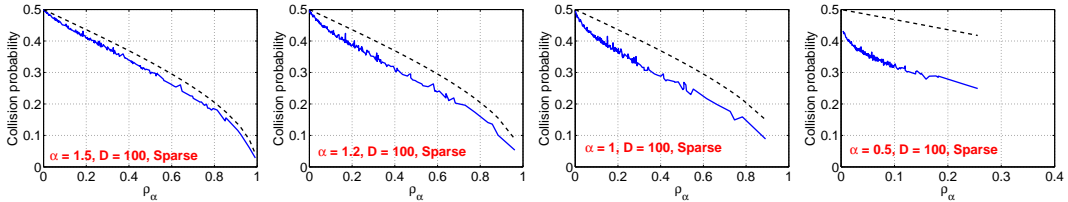

Figure 3: **Sparse Data** and $D = 100$. Simulated collision probability $\mathbf{Pr}\left(\text{sign}(x) \neq \text{sign}(y)\right)$ for sign stable random projection. The upper bound is not tight especially when $\alpha \leq 1$.

In summary, the collision probability bound: $\mathbf{Pr}\left(\text{sign}(x) \neq \text{sign}(y)\right) \leq \frac{1}{\pi}\cos^{-1}\rho_\alpha$ is fairly sharp when $\alpha$ is close to 2 (e.g., $\alpha \geq 1.5$). However, for $\alpha \leq 1$, a better approximation is needed.

## 4 $\alpha = 1$ and Chi-Square ($\chi^2$) Similarity

In this section, we focus on nonnegative data ($u_i \geq 0, v_i \geq 0$) and $\alpha = 1$. This case is important in practice. For example, we can view the data ($u_i, v_i$) as empirical probabilities, which are common when data are generated from histograms (as popular in NLP and vision) [4, 10, 13, 2, 28, 27, 26].

In this context, we always normalize the data, i.e., $\sum_{i=1}^{D} u_i = \sum_{i=1}^{D} v_i = 1$. Theorem 1 implies

$$\mathbf{Pr}\left(\text{sign}(x) \neq \text{sign}(y)\right) \leq \frac{1}{\pi}\cos^{-1}\rho_1, \quad \text{where } \rho_1 = \left(\sum_{i=1}^{D} u_i^{1/2} v_i^{1/2}\right)^2 \qquad (7)$$

While the bound is not tight, interestingly, the collision probability can be related to the $\chi^2$ similarity.

Recall the definitions of the chi-square distance $d_{\chi^2} = \sum_{i=1}^{D} \frac{(u_i - v_i)^2}{u_i + v_i}$ and the chi-square similarity $\rho_{\chi^2} = 1 - \frac{1}{2}d_{\chi^2} = \sum_{i=1}^{D} \frac{2u_i v_i}{u_i + v_i}$. In this context, we should view $\frac{0}{0} = 0$.

**Lemma 2** *Assume $u_i \geq 0$, $v_i \geq 0$, $\sum_{i=1}^{D} u_i = 1$, $\sum_{i=1}^{D} v_i = 1$. Then*

$$\rho_{\chi^2} = \sum_{i=1}^{D} \frac{2u_i v_i}{u_i + v_i} \geq \rho_1 = \left( \sum_{i=1}^{D} u_i^{1/2} v_i^{1/2} \right)^2 \square \tag{8}$$

It is known that the $\chi^2$-kernel is PD [10]. Consequently, we know the acos-$\chi^2$-kernel is also PD.

**Lemma 3** *The kernel defined as $K(u,v) = 1 - \frac{1}{\pi} \cos^{-1} \rho_{\chi^2}$ is positive definite (PD).* $\qquad\square$

The remaining question is how to connect *Cauchy random projections* with the $\chi^2$ similarity.

## 5  Two Approximations of Collision Probability for Sign Cauchy Projections

It is a difficult problem to derive the collision probability of sign Cauchy projections if we would like to express the probability only in terms of certain summary statistics (e.g., some distance). Our **first** observation is that the collision probability can be well approximated using the $\chi^2$ similarity:

$$\mathbf{Pr}\left(\text{sign}(x) \neq \text{sign}(y)\right) \approx P_{\chi^2(1)} = \frac{1}{\pi} \cos^{-1}\left(\rho_{\chi^2}\right) \tag{9}$$

Figure 4 shows this approximation is better than $\frac{1}{\pi} \cos^{-1}(\rho_1)$. Particularly, in sparse data, the approximation $\frac{1}{\pi} \cos^{-1}(\rho_{\chi^2})$ is very accurate (except when $\rho_{\chi^2}$ is close to 1), while the bound $\frac{1}{\pi} \cos^{-1}(\rho_1)$ is not sharp (and the curve is not smooth in $\rho_1$).

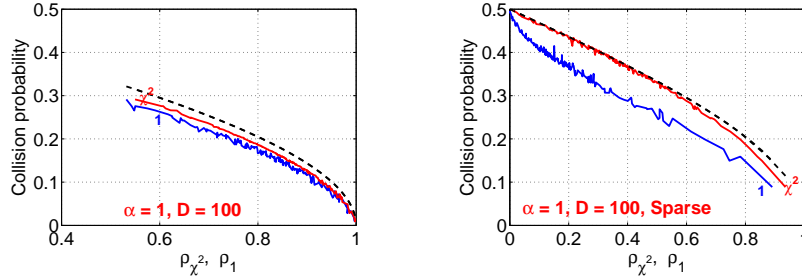

Figure 4: The dashed curve is $\frac{1}{\pi} \cos^{-1}(\rho)$, where $\rho$ can be $\rho_1$ or $\rho_{\chi^2}$ depending on the context. In each panel, the two solid curves are the empirical collision probabilities in terms of $\rho_1$ (labeled by "1") or $\rho_{\chi^2}$ (labeled by "$\chi^2$"). It is clear that the proposed approximation $\frac{1}{\pi} \cos^{-1} \rho_{\chi^2}$ in (9) is more tight than the upper bound $\frac{1}{\pi} \cos^{-1} \rho_1$, especially so in sparse data.

Our **second** (and less obvious) approximation is the following integral:

$$\mathbf{Pr}\left(\text{sign}(x) \neq \text{sign}(y)\right) \approx P_{\chi^2(2)} = \frac{1}{2} - \frac{2}{\pi^2} \int_0^{\pi/2} \tan^{-1}\left( \frac{\rho_{\chi^2}}{2 - 2\rho_{\chi^2}} \tan t \right) dt \tag{10}$$

Figure 5 illustrates that, for dense data, the second approximation (10) is more accurate than the first (9). The second approximation (10) is also accurate for sparse data. Both approximations, $P_{\chi^2(1)}$ and $P_{\chi^2(2)}$, are monotone functions of $\rho_{\chi^2}$. In practice, we often do not need the $\rho_{\chi^2}$ values explicitly because it often suffices if the collision probability is a monotone function of the similarity.

### 5.1  Binary Data

Interestingly, when the data are binary (before normalization), we can compute the collision probability exactly, which allows us to analytically assess the accuracy of the approximations. In fact, this case inspired us to propose the second approximation (10), which is otherwise not intuitive.

For convenience, we define $a = |I_a|$, $b = |I_b|$, $c = |I_c|$, where

$$I_a = \{i | u_i > 0, v_i = 0\}, \qquad I_b = \{i | v_i > 0, u_i = 0\}, \qquad I_c = \{i | u_i > 0, v_i > 0\}, \tag{11}$$

Assume binary data (before normalization, i.e., sum to one). That is,

$$u_i = \frac{1}{|I_a| + |I_c|} = \frac{1}{a + c}, \quad \forall i \in I_a \cup I_c, \qquad v_i = \frac{1}{|I_b| + |I_c|} = \frac{1}{b + c}, \quad \forall i \in I_b \cup I_c \tag{12}$$

The chi-square similarity $\rho_{\chi^2}$ becomes $\rho_{\chi^2} = \sum_{i=1}^{D} \frac{2u_i v_i}{u_i + v_i} = \frac{2c}{a+b+2c}$ and hence $\frac{\rho_{\chi^2}}{2 - 2\rho_{\chi^2}} = \frac{c}{a+b}$.

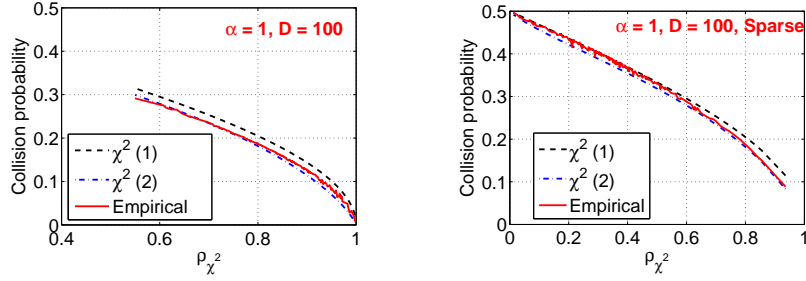

Figure 5: Comparison of two approximations: $\chi^2(1)$ based on (9) and $\chi^2(2)$ based on (10). The solid curves (empirical probabilities expressed in terms of $\rho_{\chi^2}$) are the same solid curves labeled "$\chi^2$" in Figure 4. The left panel shows that the second approximation (10) is more accurate in dense data. The right panel illustrate that both approximations are accurate in sparse data. (9) is slightly more accurate at small $\rho_{\chi^2}$ and (10) is more accurate at $\rho_{\chi^2}$ close to 1.

**Theorem 2** *Assume binary data. When $\alpha = 1$, the exact collision probability is*

$$\mathbf{Pr}\left(sign(x) \neq sign(y)\right) = \frac{1}{2} - \frac{2}{\pi^2} E\left\{\tan^{-1}\left(\frac{c}{a}|R|\right)\tan^{-1}\left(\frac{c}{b}|R|\right)\right\} \tag{13}$$

*where $R$ is a standard Cauchy random variable.* □

When $a = 0$ or $b = 0$, we have $E\left\{\tan^{-1}\left(\frac{c}{a}|R|\right)\tan^{-1}\left(\frac{c}{b}|R|\right)\right\} = \frac{\pi}{2}E\left\{\tan^{-1}\left(\frac{c}{a+b}|R|\right)\right\}$. This observation inspires us to propose the approximation (10):

$$P_{\chi^2(2)} = \frac{1}{2} - \frac{1}{\pi}E\left\{\tan^{-1}\left(\frac{c}{a+b}|R|\right)\right\} = \frac{1}{2} - \frac{2}{\pi^2}\int_0^{\pi/2}\tan^{-1}\left(\frac{c}{a+b}\tan t\right)dt$$

To validate this approximation for binary data, we study the difference between (13) and (10), i.e.,

$$Z(a/c, b/c) = Err = \mathbf{Pr}\left(\text{sign}(x) \neq \text{sign}(y)\right) - P_{\chi^2(2)}$$
$$= -\frac{2}{\pi^2}E\left\{\tan^{-1}\left(\frac{1}{a/c}|R|\right)\tan^{-1}\left(\frac{1}{b/c}|R|\right)\right\} + \frac{1}{\pi}E\left\{\tan^{-1}\left(\frac{1}{a/c+b/c}|R|\right)\right\} \tag{14}$$

(14) can be easily computed by simulations. Figure 6 confirms that the errors are larger than zero and very small . The maximum error is smaller than 0.0192, as proved in Lemma 4.

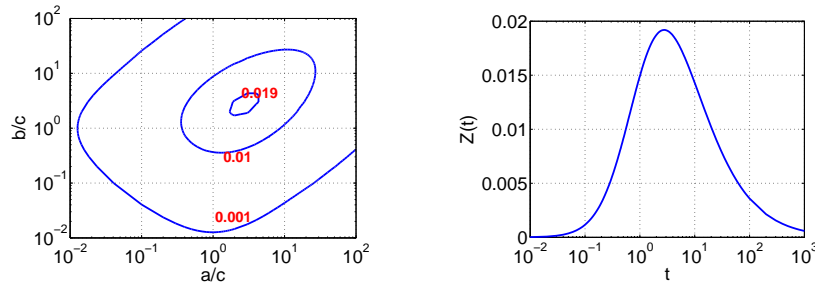

Figure 6: Left panel: contour plot for the error $Z(a/c, b/c)$ in (14). The maximum error (which is $< 0.0192$) occurs along the diagonal line. Right panel: the diagonal curve of $Z(a/c, b/c)$.

**Lemma 4** *The error defined in (14) ranges between 0 and $Z(t^*)$:*

$$0 \leq Z(a/c, b/c) \leq Z(t^*) = \int_0^\infty \left\{-\frac{2}{\pi^2}\left(\tan^{-1}\left(\frac{r}{t^*}\right)\right)^2 + \frac{1}{\pi}\tan^{-1}\left(\frac{r}{2t^*}\right)\right\}\frac{2}{\pi}\frac{1}{1+r^2}dr \tag{15}$$

*where $t^* = 2.77935$ is the solution to $\frac{1}{t^2-1}\log\frac{2t}{1+t} = \frac{\log(2t)}{(2t)^2-1}$. Numerically, $Z(t^*) = 0.01919$.* □

## 5.2 An Experiment Based on 3.6 Million English Word Pairs

To further validate the two $\chi^2$ approximations (in non-binary data), we experiment with a word occurrences dataset (which is an example of histogram data) from a chunk of $D = 2^{16}$ web crawl documents. There are in total 2,702 words, i.e., 2,702 vectors and 3,649,051 word pairs. The entries of a vector are the occurrences of the word. This is a typical sparse, non-binary dataset. Interestingly, the errors of the collision probabilities based on two $\chi^2$ approximations are still very small. To report the results, we apply sign Cauchy random projections $10^7$ times to evaluate the approximation errors of (9) and (10). The results, as presented in Figure 7, again confirm that the upper bound $\frac{1}{\pi}\cos^{-1}\rho_1$ is not tight and both $\chi^2$ approximations, $P_{\chi^2(1)}$ and $P_{\chi^2(2)}$, are accurate.

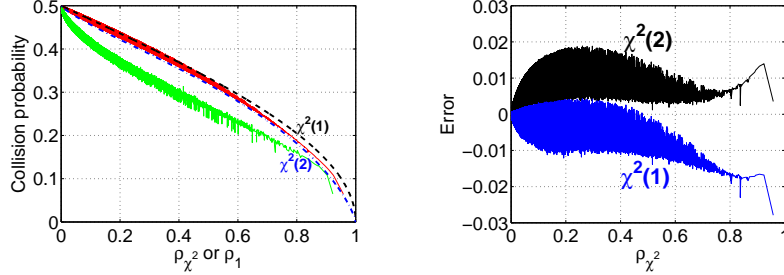

Figure 7: Empirical collision probabilities for 3.6 million English word pairs. In the **left panel**, we plot the empirical collision probabilities against $\rho_1$ (lower, green if color is available) and $\rho_{\chi^2}$ (higher, red). The curves confirm that the bound $\frac{1}{\pi}\cos^{-1}\rho_1$ is not tight (and the curve is not smooth). We plot the two $\chi^2$ approximations as dashed curves which largely match the empirical probabilities plotted against $\rho_{\chi^2}$, confirming that the $\chi^2$ approximations are good. For smaller $\rho_{\chi^2}$ values, the first approximation $P_{\chi^2(1)}$ is slightly more accurate. For larger $\rho_{\chi^2}$ values, the second approximation $P_{\chi^2(2)}$ is more accurate. In the **right panel**, we plot the errors for both $P_{\chi^2(1)}$ and $P_{\chi^2(2)}$.

## 6 Sign Cauchy Random Projections for Classification

Our method provides an effective strategy for classification. For each (high-dimensional) data vector, using $k$ sign Cauchy projections, we encode a negative sign as "01" and a positive as "10" (i.e., a vector of length 2) and concatenate $k$ short vectors to form a new feature vector of length $2k$. We then feed the new data into a linear classifier (e.g., LIBLINEAR). Interestingly, this linear classifier approximates a nonlinear kernel classifier based on acos-$\chi^2$-kernel: $K(u,v) = 1 - \frac{1}{\pi}\cos^{-1}\rho_{\chi^2}$. See Figure 8 for the experiments on the same two datasets in Figure 1: *UCI-PEMS* and *MNIST-Small*.

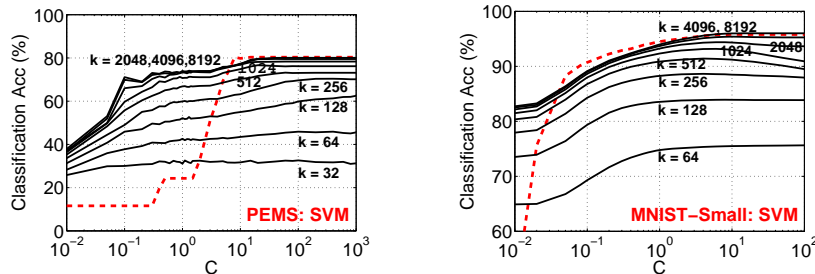

Figure 8: The two dashed (red if color is available) curves are the classification results obtained using "acos-$\chi^2$-kernel" via the "precomputed kernel" functionality in LIBSVM. The solid (black) curves are the accuracies using $k$ sign Cauchy projections and LIBLINEAR. The results confirm that the linear kernel from sign Cauchy projections can approximate the nonlinear acos-$\chi^2$-kernel.

Figure 1 has already shown that, for the *UCI-PEMS* dataset, the $\chi^2$-kernel ($\rho_{\chi^2}$) can produce noticeably better classification results than the acos-$\chi^2$-kernel ($1 - \frac{1}{\pi}\cos^{-1}\rho_{\chi^2}$). Although our method does not directly approximate $\rho_{\chi^2}$, we can still estimate $\rho_{\chi^2}$ by assuming the collision probability is exactly $\mathbf{Pr}\left(\text{sign}(x) \neq \text{sign}(y)\right) = \frac{1}{\pi}\cos^{-1}\rho_{\chi^2}$ and then we can feed the estimated $\rho_{\chi^2}$ values into LIBSVM "precomputed kernel" for classification. Figure 9 verifies that this method can also approximate the $\chi^2$ kernel with enough projections.

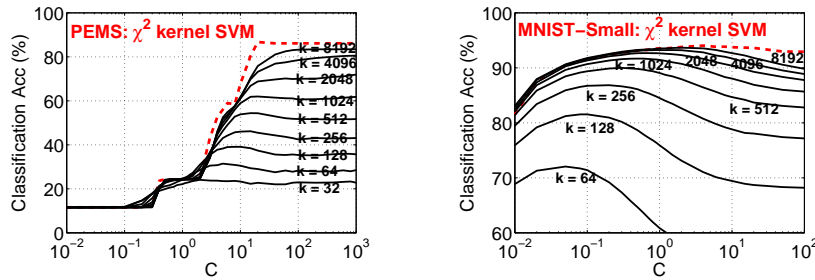

Figure 9: **Nonlinear kernels**. The dashed curves are the classification results obtained using $\chi^2$-kernel and LIBSVM "precomputed kernel" functionality. We apply $k$ sign Cauchy projections and estimate $\rho_{\chi^2}$ assuming the collision probability is exactly $\frac{1}{\pi}\cos^{-1}\rho_{\chi^2}$ and then feed the estimated $\rho_{\chi^2}$ into LIBSVM again using the "precomputed kernel" functionality.

## 7 Conclusion

The use of $\chi^2$ similarity is widespread in machine learning, especially when features are generated from histograms, as common in natural language processing and computer vision. Many prior studies [4, 10, 13, 2, 28, 27, 26] have shown the advantage of using $\chi^2$ similarity compared to other measures such as $l_2$ distance. However, for large-scale applications with ultra-high-dimensional datasets, using $\chi^2$ similarity becomes challenging for practical reasons. Simply storing (and maneuvering) all the high-dimensional features would be difficult if there are a large number of observations. Computing all pairwise $\chi^2$ similarities can be time-consuming and in fact we usually can not materialize an all-pairwise similarity matrix even if there are merely $10^6$ data points. Furthermore, the $\chi^2$ similarity is nonlinear, making it difficult to take advantage of modern linear algorithms which are known to be very efficient, e.g., [14, 25, 6, 3]. When data are generated in a streaming fashion, computing $\chi^2$ similarities without storing the original data will be even more challenging.

The method of $\alpha$-*stable random projections* $(0 < \alpha \le 2)$ [11, 17] is popular for efficiently computing the $l_\alpha$ distances in massive (streaming) data. We propose *sign stable random projections* by storing only the signs (i.e., 1-bit) of the projected data. Obviously, the saving in storage would be a significant advantage. Also, these bits offer the indexing capability which allows efficient search. For example, we can build hash tables using the bits to achieve sublinear time near neighbor search (although this paper does not focus on near neighbor search). We can also build efficient linear classifiers using these bits, for large-scale high-dimensional machine learning applications.

A crucial task in analyzing sign stable random projections is to study the probability of collision (i.e., when the two signs differ). We derive a theoretical bound of the collision probability which is exact when $\alpha = 2$. The bound is fairly sharp for $\alpha$ close to 2. For $\alpha = 1$ (i.e., Cauchy random projections), we find the $\chi^2$ approximation is significantly more accurate. In addition, for binary data, we analytically show that the errors from using the $\chi^2$ approximation are less than 0.0192. Experiments on real and simulated data confirm that our proposed $\chi^2$ approximations are very accurate.

We are enthusiastic about the practicality of *sign stable projections* in learning and search applications. The previous idea of using the signs from *normal random projections* has been widely adopted in practice, for approximating correlations. Given the widespread use of the $\chi^2$ similarity and the simplicity of our method, we expect the proposed method will be adopted by practitioners.

**Future research** Many interesting future research topics can be studied. (i) The processing cost of conducting stable random projections can be dramatically reduced by *very sparse stable random projections* [16]. This will make our proposed method even more practical. (ii) We can try to utilize more than just 1-bit of the projected data, i.e., we can study the general coding problem [19]. (iii) Another interesting research would be to study the use of sign stable projections for sparse signal recovery (Compressed Sensing) with stable distributions [21]. (iv) When $\alpha \to 0$, the collision probability becomes $\mathbf{Pr}\left(\text{sign}(x) \ne \text{sign}(y)\right) = \frac{1}{2} - \frac{1}{2}Resemblance$, which provides an elegant mechanism for computing resemblance (of the binary-quantized data) in sparse data streams.

**Acknowledgement** The work of Ping Li is supported by NSF-III-1360971, NSF-Bigdata-1419210, ONR-N00014-13-1-0764, and AFOSR-FA9550-13-1-0137. The work of Gennady Samorodnitsky is supported by ARO-W911NF-12-10385.

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
