[Reviews · NeurIPS 2013]

Submitted by Assigned_Reviewer_7

The paper studies the collision probability of the following hashing scheme for points in R^D: the hash function picks a random vector of D i.i.d. samples from a symmetric alpha-stable distribution, and to hash a point x, we compute the dot product of x and the random vector and return the sign of the dot product. The same scheme for 2-stable distribution has been studied before (known as sim-hash). The main result shows that the collision probability for alpha=1 on binary data can be approximated by a function of the chi-square similarity. The bound for general data is pointed out by the authors themselves to be far from the true collision probability so it is not clear what it means, especially regarding the comparison between that bound and the chi square similarity.

The paper provides some experiments to show that the collision probability is approximately equal to some functions of the chi square similarity, and thus one can approximate chi square similarity from the collision probability and these functions. However, in this process, I think we are no longer able to use linear SVM or efficient near neighbor search (advantages 2 and 3 in the introduction) and have to use kernel SVM and exhaustive search instead. When using linear SVM, we use the kernel implicitly defined by the LSH and there is inadequate explanation of how useful it is (chi square is useful but it does not mean any function of it is also useful).

I have not verified all the proofs in the appendix but the ones I have read seem correct. The proof details are clear and easy to follow. The question of finding a hashing scheme approximating the chi square similarity is interesting, but the paper does not get close to resolving it.

============

Rebuttal comments: the streaming application is interesting but in all 3 applications (linear SVM, near neighbor search, streaming), the crucial factor is whether the kernel implicitly defined by the hashing scheme is useful, which is still inadequately supported. It would be helpful to report the accuracy and speed achieved using the hashing scheme directly with linear SVM (as described in the introduction) on, say, the MNIST data.
Summary: The paper shows the collision probability of a certain hashing scheme based on stable distribution is related to some function of the chi square similarity. The paper does not show how useful this function is for learning tasks and how to get algorithms for chi square similarity.

Submitted by Assigned_Reviewer_8

This paper explores an novel hashing scheme that relates the chi-squared similarity between two high-dimensional vectors x and y, and the Hamming distance between the sign vectors of their randomly projected versions, sign(xR) and sign(yR). Here R is chosen to be a projection matrix whose elements are drawn from an alpha-stable distribution. The authors theoretically demonstrate that the chi-squared similarity measure is lower-bounded by the cosine of the (normalized) Hamming distance between the sign vectors. Moreover, for the case where alpha=1 and the matrix is i.i.d. Cauchy, they empirically demonstrate this bound is a good approximation to the actual similarity measures. The authors utilize this relation to achieve competitive performance with existing kernel-based methods for classification.

The ideas introduced in this paper are interesting. As the authors point out in the conclusions, the chi-squared similarity measure potentially lends itself to several important applications and therefore, a method that efficiently computes this measure can be very beneficial. I have a few concerns that the authors can hopefully address:

- The paper can benefit from some careful editing. Most of the paper concerns itself with carefully bounding the collision probabilities of the dimensionality reduced representations. However, it looks like the true focus instead (especially in practice) is the chi-squared similarity measure and this message is not really hammered home until Section 7. The experimental results are also scattered and this impedes the flow of the paper somewhat.

- A feature of most hashing schemes in this line of work (such as sim-hash, LSH, etc) is that it is possible to provide concrete lower as well as upper bound-type relations between pairwise metrics measured in the original high-dimensional space and the Hamming space. Therefore, one cannot really describe this scheme as a rigorous similarity estimation technique in the vein of the previous approaches. It would be nice if the authors comment upon the conceptual difficulties involved in proving the other half of the problem.

- While the approach presented in the paper is space-efficient, the authors do not also explicitly address the time efficiency for the case alpha = 1 where the random projection matrix is Cauchy (for alpha = 2, it is known that there exist fast transforms that achieve the dimensionality reduction).

- Lastly, it would be good if the authors normalized the x-axis to the same limits in Figs 2 and 3. It is difficult to visualize the empirical tightness of the proposed bounds when the scales of visualization are very different.

--------

Post-rebuttal edit: I would leave my score unchanged. My opinion of the paper continues to be positive, since the ideas are interesting, novel, and might have considerable potential impact. However, I echo several concerns of the other reviewers: (i) The papers needs a careful and thorough re-organization. (ii) The paper needs to address the (fairly substantial) gaps in the narrative -- particularly, the relationship between the chi-squared similarity and the approximations proposed in Theorem 2 and Lemma 5. (iii) The experimental results need to be re-organized, explained clearly, and made more convincing to a machine learning practitioner. I hope that the authors can address some of these in future revisions.
Summary: This is an interesting paper that proposes an efficient method to estimate the chi-squared similarity between high-dimensional vectors via the method of measuring Hamming distances between the signs of their alpha-stable random projections. The paper falls short of establishing a fully rigorous metric equivalence between the original space and its low-dimensional embedding, but the (fairly extensive) empirical evaluations offsets this drawback.

Submitted by Assigned_Reviewer_9

In this paper, the authors propose fast approaches to compute non-linear kernels between non-negative vectors. They focus in particular on chi_2 kernels.

Building upon the known fact that the dot-product < x, y > of two vectors can be recovered as the cosine of the collision probability P(sign(x_j) \ne sign(y_j)) where x_j and y_j are equal to the dot product of x and y with a same vector a_j whose coordinates are drawn randomly from a stable distribution with scale alpha=2. The topic of the paper is to check what happens when the scale alpha < 2 and see whether one can recover interesting approximations of other kernels. After a motivating example in Section 2 and preliminary results in Section 3, the authors introduce their real aim in Section 4 and provide in Section 5 two formulas to approximate the collision probability as a function of the chi2 similarity.

I have liked this paper, and I think it contains sufficient novel material to be a worthwhile addition to NIPS, but I have a few concerns which prevent me from being more positive about it. Basically, the paper is messy. It feels as if the authors themselves have not digested the content they propose and are still hesitating about which form the paper should take. Some expressions such as "(perhaps surprising)", and the simplicity of some of the results (upper bounds) suggest that this is still a work in progress. This makes the paper difficult to read. Here are a few examples:

= the authors motivate their interest for the chi2 similarity by fact that it's popular to compare histograms. In that case why consider time series (UCI-PEMS) of non-normalized measurements as one of their main examples? This is confusing. Besides, that dataset itself is not that large (in the context of what the authors are trying to show) since it only contains 440 points. Finally, the reader will have a hard time trying to understand why the "acos-\chi_2" kernel is introduced in l.82 without more context. I'd suggest to drop section 2 and focus more cleanly on experimental results at the end of the paper. As a side note, the accuracies on MNIST seem very low... I wonder what a Hellinger kernel or a Gaussian kernel would do.

= the whole paper is constructed to show how the chi2 similarity can be used to approximate the collision probability. However, if I understand correctly, the natural flow in applications goes the other way round: empirical collision probabilities can be computed, and hence chi2 similarities can be approximated. In that sense, I wonder whether the second approximation of Equation 12 has any sense in machine learning, unless there's an obvious inversion formula for arctan integral.

= the paper has reads like a hack in progress. Maybe the explicit layout of an algorithm at the end of what the authors think is their more important contribution could help the paper be more understandable.

= Lemma 4 is provided without a proof, and I don't think one can proceed by analogy with the proof of Lemma 2, as suggested by the authors). In the case of Lemma 2 the correspondence between features and feature products (collisions) / the kernel 1-1/pi acos(p_2) was, when taking the expectation, exact. There is not such an exact formula in the case of p_\chi_2, as stated repeatedly by the authors. Therefore, one needs to prove that, because \rho_\chi_2 is a psd kernel, 1-1/\pi acos \rho_\chi_2 is psd too. This is not "trivial" but must be proved. Using corollary 1.14 chapter 3 (p.70) of the Berg/Christensen/Ressel reference (Harmonic Analysis on Semigroups) should suffice, since we only need to show that acos is holomorphic and with positive coefficients. The following reference might be a good starting point:
http://mizugadro.mydns.jp/t/index.php/ArcCos#Range_of_holomorphism
At this point, however, lemma 4 is just a conjecture.

update after rebuttal:

The answer from the authors on the psdness of the acos chi2 did not convince me. I don't understand their proof. The feature representations for the chi2 kernel are, as far as I know, infinite dimensional. Then the authors propose to use random Gaussian approximations on infinite dimensional vectors? I am sorry, I can't follow this. This part needs to be proved more clearly.
Summary: This nice paper reads like a mishmash of ideas that try to generalize simhash (alpha=2) for stable distributions with alpha < 2 and see what can be taken out of it. Some of these ideas are promising, but the presentation needs to be improved to be suitable for NIPS

Submitted by Assigned_Reviewer_10

This paper shows that the collision probability after alpha-stable random projections can be approximated by functions of Chi-Sqaure similarity. In particular, when alpha = 1, the approximation has a close form. This result enables approximating Chi-Sqaure related kernels via linear kernels after sign Cauchy random projection.

However, the bounds can be loose, and the application part seems rush and incomplete at the moment.

The paper can be (easily) improved by turning into a more theoretical paper with only hints for applications. After all, the main contribution is the theoretical part. Despite the looseness of the bounds and the issue of the current application part, the paper is interesting, and may still have values in practice or at least motivate new techniques.

After reading the reviews, author response, and discussion, my score (7) remains unchanged. I prefer a novel though unpolished paper to a polished incremental paper. I hope the writing will be improved in the final version if accepted.

Summary: The result is original. The new bounds are useful addition to the literature and the community.

Submitted by Assigned_Reviewer_11

This paper proposes to use a signed Cauchy random projection to approximate the chi-square kernel with relatively low errors. The motivation comes from generalizing the Sim-hash algorithm and noting that chi-square is related to the collision probability.

Overall, the reviewer finds the finding quite interesting and innovative, as no previous attempts seem to have linked chi-square with random Cauchy projections. The paper is of importance to very high-dimensional sparse approximations of the chi-square, which could find its usage in a lot of text and image applications.

One main concern is that the writing is very bad, almost atrociously so. From the beginning it is extremely unclear what is the goal of the paper. The first theorem is almost irrelevant with the rest, as it gives a bound, just to be said not tight. And starting from section 4, the focus has turned to the alpha=1 case where links between the collision probability and the chi-square is proposed, and later proved in the binary case. The logic seems completely reversed and free-flowing, with occasionally theorems that don't lead to anywhere (e.g. Theorem 1, and the approximation (12)). The claims in the conclusion look out-or-order and random (e.g. lines 424-427). In general, this looks more like a research memo rather than a rigorous, well-organized paper. Therefore I'm torn on a score between 5 and 6 since it looks like this paper should be completely re-written before presented to readers.

However, the finding itself is still quite interesting. Lemma 5 is a strong result that established a low error bound on the approximation in binary case. The result that one can approximate cos^{-1}(chi^2) by inner product is new. A less satisfying result is the way chi^2 is approximated, where cos needs to be applied to transform sign(x)* sign(y) so that the approximation is no longer an inner product thus don't enjoy computational savings in large-scale learning. It will be interesting to see whether the approximated cos^{-1}(chi^2) can translate to a linear approximation to the more useful chi2 CDF function, or whether some more theoretical and empirical results can be found for the cos^{-1}(chi^2).
Other notes and relevant references:

Note that when alpha = 1, the bound in Theorem 1 corresponds to the geodesic distance on the simplex, as defined in

Lafferty and Lebanon. Diffusion Kernels on Statistical Manifolds. JMLR 2005.

Taking the square root of all the input u, v is known as the Hellinger distance, widely used in both NLP and vision but widely known as inferior to chi-square.

Chi-square is very important at least in computer vision, quite a few papers have worked on its approximation. There are some additional references that might be interesting for this topic:

Sreekanth, Vedaldi and Zisserman. Generalized RBF feature maps for efficient detection. BMVC 2010

established an approximation for exp-chi2, which has been empirically verified as much better than the chi2 kernel in classification accuracy. Also note exp-chi2 resembles the CDF of the chi2 distribution, hence corresponds to the p-value of the chi2-test. Therefore it's more appropriate than the chi2 itself as a comparison metric. Also relevant is a recent arXiv paper provides a linear approximation to chi2 with geometric convergence rate:

Li, Lebanon and Sminchisescu. A Linear Approximation to the chi2 Kernel with Geometric Convergence.

http://arxiv.org/pdf/1206.4074v3.pdf

which presents a formula to approximate each dimension of the chi2 kernel with several dimensions with geometric convergence rate. One can in principle construct an approximation of chi2 on very high-dimensional sparse features by doing a standard Gaussian random projection from such an approximation. It would be interesting to compare these two approaches.

---------------------------------------------------------------------

Comments on Rebuttal:

The point the authors mentioned on the ability for applications in streaming data is correct and interesting, and give some merit to the otherwise unjustified acos(chi2) metric (since it's computable for streams, even if it's not a well-understood metric yet). Therefore I decide to change the score to 6. The main concerns are still 1) Lack of theoretical and/or practical insights of the acos(chi2) kernel. 2) Writing and organization of the paper.

Performance comparisons with the arXiv paper would be an interesting side note, but that was not a factor in the score I gave with the initial review. The score came from the above 2 concerns.
Summary: The paper proposes an approach to approximate the cos^{-1}(chi2) kernel using signed random Cauchy projections. The approach is interesting but it's unclear what theoretical/empirical meanings do the approximated cos^{-1}(chi2) have. The writing is extremely messy which undermines the score of this paper.
Author Feedback

Author rebuttal: Reviewer_9:

(1) Lemma 4 is indeed analogous to Lemma 2. Because \rho_{chi2} is PD, we can write it as \rho_{chi2} = sum_i z_i w_i, even though we don’t know z, w explicitly. When two original data vectors are identical, we have \rho_{chi2} = 1, which means sum z_i^2 = sum w_i^2 = 1 (ie naturally normalized). At this point, we can apply normal random projections and the rest follows from Lemma 2. There is another perspective: Since \rho_{chi2} is PD, there exists a Gaussian Process with covariance \rho_{chi2}. The variance turns out to be 1 = sum_i u_i ; and the rest follows easily. We of course appreciate your kind suggestion about holomorphic functions and we did learn something useful from reading the link. Also, we agree it always helps readers if we could provide more details (if space allows).

(2) In practice, we usually don’t explicitly need numerical numbers of similarities. Often we just need a good smooth function which is monotone in similarity. The second approximation is motivated by the analysis of collision probability on binary data, which is the case we can fully analyze.

(3) For MNIST-small, we are actually impressed by the performance of chi-square kernel compared to linear kernel. Note that we only used 1/6 of the original training samples and the entire original testing samples. Linear SVM achieved 90% while chi-square kernels achieved about 96%. Like linear kernel, chi-square kernel does not have additional tuning parameter (like gamma in RBF). This is actually the example which makes us feel excited about chi-square kernels. And Yes, MNIST is more close to vision data than PEMS. For this paper we used them mainly for verifying the proposed approximations. We realize we were a bit too enthusiastic about linear SVM and we agree with you if we add additional tuning parameters (similar to RBF), the accuracy will be further improved. In industry practice (where the data are not as nice as MNIST), it appears that linear kernels are the most common. The recent blog article "Research Directions for Machine Learning and Algorithms" (May 16, 2011) in the Communications of the ACM (CACM) confirmed this industry practice as well.

(4) In addition, we should mention another significant advantage of this method. It naturally works with streaming data. The process of histogram constructions is a typical example of Turnstile data streams. Using this method, we don’t need to store the original massive data.
Thanks for the careful reading and thoughtful constructive suggestions.

Reviewer_8:

(1) Speedup. Yes, one can improve the processing cost for random projections easily by “very sparse stable random projections” (Li, KDD07). The projection matrix can be made extremely sparse, especially for \alpha = 1 (see the experiments in KDD07 paper).

(2)x-axis. If we normalize the x-axis to the same scale, then we won’t see anything for \alpha=0.2. As explained in the paper, the experiments cannot cover the entire range of \rho_\alpha. We will try a few things to see which provides the best compromise.
Thanks for pointing this out.

(3) If the collision probability is exact, then the inequality relations (as the results people usually present in LSH type of papers) immediately hold. The key in LSH is that the collision probability should be a monotone function of the similarity. Both approximations are monotone functions of \rho_chi2.
Thank you and other reviewers for pointing out the editing.


Reviewer_11:

Thanks for confirming the high importance of chi-square similarity in practice. Note that our method works with data streams (Turnstile model). Histograms are naturally data streams. The other methods you mentioned require storing the original data at some step. Denote the original data matrix by A, then random projection A*R can be computed incrementally since (A+dA)*R = A*R + dA*R, meaning that A is never stored (R is re-generated on-demand by pseudo-rand numbers). This is a significant advantage of linear projections. The methods you mentioned, e.g., arXiv:1206.4074, first applied a nonlinear operation on A and expanded the dimension by a factor of 5 to 10, before random projections. Consequently they can are not applicable to data streams. This is a big difference. Also note that our method is very simple. Again, thanks for the references and other suggestions.

Reviewer_7:

(1) The bound (Theorem 1) is rigorous and sharp when \alpha>1.5. In machine learning practice, \alpha can be viewed as tuning parameter. Our contribution is not limited to \alpha=1.

(2) The use of acos(\rho_2) (sim-hash) is standard practice in search and learning when using sign normal random projections (which is a very influential work). For near neighbor search, it is actually difficult to find alternative method which works a lot better than sim-hash. This is why we are enthusiastic about the use of acos(\rho_chi2) in practice (which can be approximated by very efficient linear kernel).

(3) For binary data, we analytically showed that the max error is <0.01919. Binary data are important in practice especially when data are high-dimensional.

(4) For non-binary data, we have extensive experiments to verify the approximations.

(5) Our method naturally works with data streams.
This is the first paper on sign stable random projections. We expect quite a few related papers (practical or theoretical) can be written in the future (by other researchers).

Reviewer_10: Thanks for suggestions. It is fun reading so many constructive comments.